# Eating Habits and Their Relationship to Oral Health

**DOI:** 10.3390/nu12092619

**Published:** 2020-08-27

**Authors:** Dennys Tenelanda-López, Pedro Valdivia-Moral, Manuel Castro-Sánchez

**Affiliations:** 1Faculty of Natural Resources, Escuela Superior Politécnica de Chimborazo, Riobamba EC060155, Ecuador; 2School of Dentistry, Universidad Nacional de Chimborazo, Riobamba 060150, Ecuador; 3Faculty of Education, Universidad de Granada, 18011 Granada, Spain; pvaldivia@ugr.es (P.V.-M.); manuelcs@ugr.es (M.C.-S.)

**Keywords:** eating habits, dental caries, DMFT index

## Abstract

The objective of this research was to compare the healthy behaviors and caries index of young people in school to obtain an overview of their lifestyles, which would enable the development of educational programs for the promotion of oral health. The study design was carried out using a descriptive, cross-sectional, and observational methodology with a mixed approach. 380 twelve-year-old students participated in this research conducted in the city of Riobamba-Ecuador. The techniques used were observational and surveys with their respective instruments, the Dental Clinical History, and the Health Behavior in School-aged Children 2014-Spain questionnaire. The community index of the Decayed, Missing due to caries, and Filled Teeth (DMFT) reflected a high level (6.47) in the study subjects. A variety of foods such as fruits, chips, vegetables, candy, sugar-containing drinks, meat, fish, dairy, and cereals were consumed at least once a week by most students. Two statistically significant associations were demonstrated in this investigation. The first one was between fruit consumption and the DMFT index, the second one was between vegetable consumption and the DMFT index. Both associations showed significant values (*p*) of 0.049 and 0.028, respectively; these were not determining indicators since caries is a multifactorial pathology, which can develop not only as a product of poor eating habits.

## 1. Introduction

The World Health Organization (WHO) considers that having good oral health is essential for having good quality of life. Oral health is conceptualized as the absence of oral-facial pain, mouth or throat cancer, infections and mouth sores, periodontal diseases, caries, tooth loss, and other diseases and disorders that limit the person to perform the functions of the stomatognathic apparatus such as the ability to bite, chew, smile and talk [1].

According to the WHO and the World Dental Federation (WDF), cited in Ortega [2], it is estimated that the prevalence of dental caries in children under 12 years of age corresponds to no more than three teeth maximum by 2020. In this sense, the mission of the Ecuadorian Ministry of Public Health is to promote strategies and actions for the prevention and promotion of oral health based on the lifestyles of its population [3].

Eating habits are understood as the set of routine actions that an individual has to feed himself; the context, the people around them, the economic availability, and the knowledge of the nutritional value of food become determining factors in forming good or bad eating habits [4]. Bad eating habits have been directly related as a risk factor for suffering from different chronic diseases, which in turn are associated with some pathologies of the oral cavity [5]. The development of caries is sometimes linked to the interaction that occurs from inadequate eating habits, the number of nutrients in food, poor oral hygiene habits, and a high level of bacterial plaque [6].

Some animal studies have shown that some foods high in fat, protein, calcium, and fluorine could have cariostatic effects. The Spanish Pediatric Society states that “In animal studies, they have observed that foods high in fat, protein, calcium, and fluorine can protect against cavities. Fats cover the tooth, reducing the retention of sugars and plaques; they can also have toxic effects on bacteria. Proteins increase the buffering capacity of saliva and have a protective effect on enamel. Together, fats and proteins raise the pH after carbohydrate intake. Another type of food would be those that, through chewing, stimulate salivary flow and, in this way, buffer the acid pH and favor the enamel remineralization” [7].

Caries is a multifactorial disease, which means that there are many risk factors that may contribute to the development of early caries in children or adolescents such as excessive consumption of sugars, bad eating habits, inadequate tooth brushing, knowledge of children and parents about oral hygiene, brushing frequency, visits to the dentist, inadequate or excess fluoride, high levels of infection because of *Streptococcus mutants* or lactobacilli, and so on [8].

University teaching must seek educational and investigative mechanisms to contribute to the achievement of the goals of the WHO in relation to the prevention of oral pathologies such as caries. An overview of the lifestyles of children or adolescents is necessary to find the most appropriate mechanisms that contribute to better current educational programs for the promotion and prevention of oral health. These programs should involve all the entities of their socio-educational community, such as primary school teachers, students, and teachers of schools of dentistry. In this way, adequate oral hygiene and feeding habits will be fostered to maintain proper oral health and comprehensive development for elementary students.

The number of Decayed, Missing due to caries, and Filled Teeth in the permanent teeth is known as the DMFT index, one of the WHO’s indicator age used to calculate it is 12 years old. There are a few studies that specifically consider this age group. So, this study is important in terms of providing the current caries level associated with one of its factors like eating habits in a population of a third-world country, considering the age group suggested by WHO [9,10].

The objective of this research is to compare the eating habits and caries index of students through the Health Behavior in School-aged Children (HBSC) [11] questionnaire in order to obtain an overview of adolescent lifestyles, which would allow developing educational programs for the prevention of general and oral disease in schoolchildren in the city of Riobamba-Ecuador. 

## 2. Materials and Methods 

The study was carried out using a descriptive, cross-sectional, and observational methodological design with a mixed approach. The study population consisted of 34,107 students legally enrolled in the Education District of the city of Riobamba-Ecuador in the academic year 2019–2020, of which, a representative sample size was calculated using a statistical formula [12], obtaining a number of 380 twelve-year-old school children, who were selected by applying a random probabilistic and stratified sample. It meant that the existing percentages of the study population were respected for getting the sample, so, there were 114 students (30%) from the rural area and 266 ones (70%) from the urban area, and according to the gender 206 (54.2%) male ones, and 174 (45.8%) female ones. The selection criteria for the students to be part of this research were being registered in public educative institutions in Riobamba, being 12 years old, having informed consent signed by their parents, knowing enough information about their parents, having permanent teeth, avoiding potable water intake, and preferring bottled/boiled water intake.

The techniques used were observational and surveys with their respective instruments, the Dental Clinical History [13], and the HBSC questionnaire [11], respectively. The information was obtained from the odontogram as the first instrument used by students in the final year of the School of Dentistry of the National University of Chimborazo, this one provided data of pieces that presented problems such as decayed, missing, and filled teeth (DMFT index) [9,14], through an intraoral examination. This index determined the high, medium, or low risk of caries in the population of study. The total value (2460) of all decayed (1744), missing (144), and filled (572) teeth pieces divided by the number of individuals examined (380) was considered to establish the DMFT community index of this research, giving a value of 6.47; the second instrument was used to obtain information on eating habits; the dimensions used were chosen from the eating habits section of items 2.3 to 2.11 of the HBSC 2014-Spain questionnaire.

The legal representatives of the students signed an informed consent for the voluntary opening of the Dental Clinical History and the application of the questionnaire to their representatives in this investigation. Additionally, children had the corresponding permission from the District Directorate of Riobamba-Ecuador. Spearman’s Rho test was used to check whether there was an association between the variable eating habits and the DMFT risk level [15]. The collected data were processed through the SPSS statistical program version 24.

## 3. Results 

The results obtained from the instruments applied to the 380 students showed that most of them were identified as male (54.2%) and the rest as female (45.8%). It was evident that more than half of the participants were at medium (27.1%) and high (40.8%) risk of the caries severity level, according to the DMFT index, it meant both risk levels added up to 67.9%. In relation to all the subjects studied, their DMFT community index was 6.47, which showed that they were at a high level of risk in general terms (Table 1).

In relation to the consumption of different foods during the week, it was observed that most people (28.2%) consumed fruits two to four days a week, on the other hand, fried potatoes and snacks were consumed by the majority (36.8%) once a week. In the range of two to four days a week, most of the students (24.2%) ate vegetables, 20.33% of the students had sweets, drinks containing sugar were consumed by 24.5%, and 27.5% were fed on meat. A total of 35.5% consumed fish once a week; the foods that were consumed every day more than once were milk or dairy (28.2) and cereals (37.6%). All the foods indicated above were consumed at least once a week by the study subjects. The foods that had never been eaten that stand out the most were the fried potatoes and salty snacks in 12.6% and the fish in 10.8% (Table 2).

Each of the foods consumed and the caries index were correlated in order to analyze their association in the context presented in this investigation; considering that caries is a multifactorial pathology, it was demonstrated that there was a statistically significant relationship between fruit consumption and DMFT risk levels with a *p*-value equal to 0.049, showing a low and positive correlation, just as it was evidenced that vegetable consumption is related to the DMFT levels of the individuals in this investigation with a *p*-value equal to 0.028, showing a low and negative correlation. Other foods such as chips and salty snacks, sweets, beverages containing sugar, meat, fish, milk or dairy products, cereals did not show a statistically significant relationship with the irrigation levels of the DMFT because they had a *p*-value equal to 0.207, 0.284, 0.426, 0.539, 0.13, 0.348, and 0.876, respectively (Table 3).

## 4. Discussion

The DMFT index is one of the most reliable indexes used to measure the level of community caries index [16].; that is why the results of the present study were obtained using these parameters. The students enrolled in this research showed a range of community caries index of 6.47, which according to the WHO is high. This result coincides with that obtained by Medina et al. [17] who determined a risk level of 5.25 in a population of the Ecuadorian Amazon that was also considered high. However, there are four studies that showed opposite results related to ours and Medina et. al. The first one was carried out by Mitta et al. [18] and showed a community caries index of 1 (low) in their population of children of Gurgaon in India. The second one evidenced that the community caries index was low (1.37) in 12-year-old children in the state of Telangana-India [19]. The third one was carried out in children from the Madinah region of Saudi Arabia, who presented a low level of community caries index (1.58) [20]. Finally, a study performed by Narbutaitė et al. also determined a low community caries index (2) in Linunia [21].

Ecuador showed that it has many shortcomings in preventing and adequately promoting oral health in its population, a problem evidenced by the high levels of caries found in the aforementioned studies in this country. Additionally, there are other possible limitations to be considered, according to the official website of the Ministry of Public Health of Ecuador; its manual called “Oral Health for Teachers and Promoters” has not been updated since 2010 nor has a national study of the state of oral health since 1996 been carried out [22]. Despite the fact that caries is a multifactorial pathology [1], this neglect of the government in the topic of oral health could explain the high level of caries in this research. The data from our study and [17] coincided with the report given by the International Dental Federation, which assigned Ecuador a value of more than 3.5 of the DMFT index (high) [23]. Other countries such as India, Saudi Arabia, and Lithuania showed a low level of caries, which could be explained by the improvement of oral health policies in their countries from 2014 to 2017, or they might be very particular cases in which local organizations that promote optimal oral health may have intervened. However, other data provided by the DMFT showed that from 1994 to 2014 there was a high rate of caries (more than 3.5 in DMFT index) in these countries [23], which disagreed with the data by [18,19,20,21], considering that it was a very short time between the mentioned events.

In this research, the eating habits of the students, two to four days a week, at risk of caries showed that they consume fruits (28.2%), vegetables (24.2%), meat (27.6%), fish (16.3%), and dairy (24.2%), percentages similar to those ones described by Doichinova et al., where in their study reflected that their population at risk of caries consumed fruit (20%), meat (39%), fish (24%), and dairy (28%); however, the consumption of vegetables (43%) is the biggest dissimilarity between this investigation and theirs [24]. It is important to highlight that in our research, the most consumed foods are milk (28.2%) and cereals (37.6%), which are consumed multiple times every day; this could be attributed to the serving of milk daily as a part of breakfast in public schools [25], plus the possibility of consuming it again at home; this agrees with Machuca et al. [26], who in their study determined that 70% of primary school children consume dairy products. On the other hand, the cereal intake would be related to data provided by the Ecuadorian Institute of Statistics and Census that states that rice is one of the most consumed foods in Ecuador [27].

Diet and eating habits can vary according to the cultural, educational, and even religious context [28]. According to authors such as Punitha et al., food consumption is linked to the risk of developing caries. In their study, the percentage obtained in individuals who consume vegetables was 31.6% of caries presence [29], which coincides with the results of our research, which showed the association of the DMFT index variable and eating habits with a significance value of 0.028, with a low and negative correlation that showed that the higher the consumption of vegetables, the lower the level of caries. These results are supported by the cariostatic properties of vegetables that contribute to maintaining adequate oral health [30]. In relation to fruit consumption, an association between this and DMFT levels is evident with a significance value of 0.049, with a scant and positive correlation that indicates that the higher the fruit consumption, the higher risk of having cavities, a result that agrees with the theory that mentions that frequent consumption of fruits, especially critical ones, generates the possibility of developing cavities [31]. In this study, there was no evidence of an association between candy consumption and caries (*p* > 0.05); however, in a study presented by Sheiham and James [32], the DMFT index caused by candy consumption was 2.5, establishing a parameter of moderate risk of its study population. Milk consumption (*p* > 0.05) did not show a statistically significant association between the variables. Even though meat, fish, and potatoes are cariostatic foods, they did not show any relationship with the caries index (*p* > 0.05), the reason being due to the low potential cariogenic of these foods. [33,34].

## 5. Conclusions

The DMFT index evidenced a high risk in most of the young people enrolled in the cities of Riobamba, and no statistical significance regarding the gender with which they identify. This index demonstrates the lack of attention of updated strategies for the promotion and prevention of oral health by state entities, and for this reason, Ecuador was classified by international dental organizations as a country with a high rate of caries in general terms.

A variety of foods such as fruits, chips, vegetables, candy, sugar-containing drinks, meat, fish, dairy, cereals are consumed at least once a week by most of the study subjects. In this research, there was only a statistical significance between the consumption of fruits and vegetables with the DMFT caries index. However, these are not determining indicators; since caries is a multifactorial pathology, which can develop not only as a product of poor eating habits. So, the future educational programs for the prevention of oral health in school children should consider planning to talk about as many caries factors as possible and not just focusing on eating habits.

## Figures and Tables

**Table 1 nutrients-12-02619-t001:** DMFT risk levels and gender.

	Male	Female	Total
**Low**			
Frequency	70	52	122
Percentage	18.4	13.7	31.1
**Medium**			
Frequency	47	56	103
Percentage	12.4	14.7	27.1
**High**			
Frequency	89	66	155
Percentage	23.4	17.4	40.8
**Total**			
Frequency	206	174	380
Percentage	54.2	45.8	100
**DMFT Community index**			6.47
**DMFT risk category:** Low ≤ 2.6, medium 2.7–4.4, high ≥ 4.5			

**Table 2 nutrients-12-02619-t002:** Eating habits frequency

	Never	Less than Once a Week	Once a Week	2–4 Days a Week	5–6 Days a Week	Once a Day, Every Day	Every Day, More than Once	Total
**Fruits consumption**								
Frequency	4	20	42	107	62	71	74	380
Percentage	1.1	5.3	11.1	28.2	16.3	18.7	19.5	100
**Chips and savory snacks consumption**								
Frequency	48	56	140	78	23	24	11	380
Percentage	12.6	14.7	36.8	20.5	6.1	6.3	2.9	100
**Vegetables or greens consumption**								
Frequency	7	22	50	92	81	55	73	380
Percentage	1.8	5.8	13.2	24.2	21.3	14.5	19.2	100
**Sweets consumption**								
Frequency	38	62	101	77	37	39	26	380
Percentage	10.0	16.3	26.6	20.3	9.7	10.3	6.8	100
**Drinks containing sugar consumption**								
Frequency	32	49	68	93	55	31	52	380
Percentage	8.4	12.9	17.9	24.5	14.5	8.2	13.7	100
**Meat consumption**								
Frequency	10	14	71	105	63	43	74	380
Percentage	2.6	3.7	18.7	27.6	16.6	11.3	19.5	100
**Fish consumption**								
Frequency	41	68	135	62	32	19	23	380
Percentage	10.8	17.9	35.5	16.3	8.4	5.0	6.1	100
**Milk and derivatives consumption**								
Frequency	5	17	26	92	61	72	107	380
Percentage	1.3	4.5	6.8	24.2	16.1	18.9	28.2	100
**Cereal consumption**								
Frequency	3	16	24	65	52	77	143	380
Percentage	0.8	4.2	6.3	17.1	13.7	20.3	37.6	100

**Table 3 nutrients-12-02619-t003:** DMFT risk levels and eating habits frequency.

	DMFT Risk Levels		
Spearman’s Rho	Correlation Coefficient	Sig. (Bilateral)	N
Fruits consumption	0.101	0.049 *	380
Chips and savory snacks consumption	−0.065	0.207	380
Vegetables or greens consumption	−0.113	0.028 *	380
Sweets consumption	0.055	0.284	380
Drinks containing sugar consumption	0.041	0.426	380
Meat consumption	−0.032	0.539	380
Fish consumption	−0.078	0.13	380
Milk and derivatives consumption	−0.048	0.348	380
Cereal consumption (rice and others)	0.008	0.876	380

Significance: * *p* < 0.05.

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
