# Peer review of "Eating Habits and Their Relationship to Oral Health"

_nutrients, 2020, doi:10.3390/nu12092619_

Round 1
Reviewer 1 Report
The authors have to be commented for taking the time to conduct this study. Studies on specific populations are important in determing the health factors associated with diseases in different cultures. However, I have concerns in regards to the importance of this study to an international audience.
The introduction should include the findings of previous studies on the topic in order to justify the missing evidence that this study will provide. The information provided in this paper is vague without any specific details on the existing knoweldge of oral health and diet of the specific population. As a result the objective of this paper is not clear. Also, some grammatical errors were noted on lines 55 and 64.
In the materials, figure 1 is not necessary as it does not provide any important information. It would be more important to describe how the authors conducted the stratification and randomization of the sample instead. Did they include children from schools of different socioeconomic areas? Were the selected children clustered?
In regards to the methodology, Dental Clinical History is not a scientifically known method so more detail and evidence of its validity is necessary. I could not open the link of ref 10 and 12 to get more details on this methods, thus I cannot be sure if the specific methodology in correct. Also, I could understand the term "pieces" that is used, are we talking about teeth, surfaces or what? Also, when were these clinical data recorded in comparison to the dietary habits, who performed the clinical examinations and using which criteria?
As I could not understand the indices used in the specific paper I cannot be sure if spearman's correlation is the correct statistical test for this study. I believe that the best way to detect if the dietary habits are related to the caries prevalence would be to compare the caries index with the diet of each subject using another statistical test.
The results presented are only descriptive without any critical thinking on the findings. Statistics may propose anything but results must present the findings that are supported from scientific logic and evidence.
Author Response
REVIEWER 1 (changes in green)
We do appreciate your suggestions that improved the article, the comments of the three reviewers were tried to be considered. Some lines changed the number because of the modifications.
In the introduction, this paragraph was added: “The number of Decayed, Missing due to caries, and Filled Teeth in the permanent teeth is known as DMFT index, the 12-year-old is one of the WHO indicator age groups to calculate it. There are a few studies that consider specifically this age groups. So, this study is important in terms that provide the current caries level associated to one of its factors like eating habits in a population of a third-world country”
Line 55 and 64 were corrected
Fig.1: It was deleted
The stratification was explained
The link of ref 10 was corrected (now, it is the link 13)
The link of ref 12 was corrected (now, it is the link 14)
It was clarified that the odontogram of the Dental Clinical History was used because it is general and very similar in different countries, and this case it is validated by the Ministry of Public Health of Ecuador.
It was corrected the paragraph in which the word “pieces” was mentioned.
The use of the Spearman’s test was justified including the information from the article of Reguant et. Al (ref: 15)

Reviewer 2 Report
Fig 1 - There is no reason to show this. It is a power calculation, it is trivial.
Page 3, Line 83 -85; I have no idea what that sentence is saying. What instrument? What dimensions? What are items 2.3 - 2.11?
Pg 3, Line 95; What does "(67, 9%)" mean?
What is a high DMF community index? What is a low? 1000? 12? 1? This is fairly general journal, the authors need to give much more background.
Why Spearman's rho? Was all the data non-parametric? Why not use multiple regressions?
Pg 6, Line 179; medium or high? Which?
Pg 6, Line 174; I do not know what irrigation means here?
'low cariogenic content of these foods" - cariogenic content is a nonsensical term. Foods do not contain "cariogens."
The term "general terms" is used so much for no reason.
Discussion is very confusing and the sentences are much too long. A sentence should never, ever be 4 or 5 lines. This is poor English.
The authors talk about the multifactorial onset of caries but never give further details.
How the authors will use this information to develop a learning program is not further explained.
Author Response
REVIEWER 2 (changes in red)
We do appreciate your suggestions that improved the article, the comments of the three reviewers were tried to be considered. Some lines changed the number because of the modifications.
Fig 1: It was deleted
Page 3, Line 83-85: It was clarified that the dimensions corresponded to the HBSC questionnaire.
Line 95: the value of 67,9% was explained that was considered, which meant most of the study population
DMF was from the abstract named for Decay, Missing and Filling Teeth
DMFT community index is ranked: Low ≤ 2.6, medium 2.7-4.4, high ≥ 4.5. This issue was included in table 1
All data was non-parametric
The use of the Spearman’s test was taken from the article of Reguant et. Al (ref: 15)
Pg. 6, Line; it was clarified as high
Pg. 6, Line 174: the translation was wrong; the correct was moderate risk
“low cariogenic content of these foods” was changed by low potential cariogenic
The multifactorial onset of caries was clarified in the introduction.

Reviewer 3 Report
ABSTRACT:
Appropriate
INTRODUCTION:
Line 62 - 66: Reference for the questionnaire is missing. Sentence formation is not appropriate.
The hypothesis of the study was not stated.
MATERIALS AND METHODS
Dental caries is associated with multifactorial etiology. Author’s didn’t mention any inclusion, exclusion criteria, Forex: Daily fluoride intake through water, or any preventive oral hygiene measures followed at home, medical condition of participants, current medications, etc.
Mention the significance of this study as many there are many studies with similar parameters.
Details on who involved in calculating DMFT from the subjects (Examiner/observers, if one intra-examiner reliability if two intra and inter-examiner reliability)
DISCUSSION:
Reference Author’s name should be followed by a year
Fluoride intake also has a very important role in dental caries and Fluoride levels in the water where the study conducted and the demographic background of the participants should be discussed to generalize the findings.
The results defiantly open a new room for researchers working on dental caries.
The authors found high caries risk in children with consumption of vegetables and fruits, however, the frequency of intake and its influence on DMFT could also be considered for evaluation.
CONCLUSIONS:
Objective-based
REFERENCES:
Most of them are web-based references/links and authors are recommended to choose journal citations if possible
Author Response
REVIEWER 3 (changes in blue)
We do appreciate your suggestions that improved the article, the comments of the three reviewers were tried to be considered. Some lines changed the number because of the modifications.
Line 62: reference was added
Line 66: sentence was connected to the previous idea.
The hypothesis was not considered because the research was focused on objectives because it was just descriptive.
MATERIALS AND METHODS:
The selecting criteria was included
It was clarified that the process was intra examiner reliability
DISCUSION:
Reference Author´s name should be followed by a year: I have read the instructions for author and some article in the Nutrients journal, but they didn´t say anything about the year, however if you consider that I have to, I will do it.
CONCLUSIONS
A part of the objective was modified in order to have a clearer conclusion objective-based
REFERENCES
Journal citations were considered, I just clarify that the recognized health organisms were considered for web-based links

Round 2
Reviewer 1 Report
Some more editing of the English language would improve the manuscript.
Author Response
We do appreciate your suggestions; they have improved the manuscript in many senses.
The spelling of complete manuscript was checked

Reviewer 2 Report
The authors have made the suggested changes.
Reviewer 3 Report
Fluoride background should be discussed
Dental caries is multi-factorial, authors should describe the findings of the study and supporting studies should be discussed.
Author Response
We do appreciate your suggestions; they have improved the manuscript in many senses.
The spelling of complete manuscript was checked
The selection criteria were established (line 84, 85) in order to exclude the possibility of caries produced by Fluoride.
